# Control of cell death/survival balance by the MET dependence receptor

Leslie Duplaquet[1], Catherine Leroy[1†], Audrey Vinchent[1†], Sonia Paget[1], Jonathan Lefebvre[1], Fabien Vanden Abeele[2], Steve Lancel[3], Florence Giffard[4], Réjane Paumelle[3], Gabriel Bidaux[5], Laurent Heliot[5], Laurent Poulain[4], Alessandro Furlan[1,5]*, David Tulasne[1]*

[1]Univ. Lille, CNRS, Institut Pasteur de Lille, UMR 8161 - M3T - Mechanisms of Tumorigenesis and Targeted Therapies, Lille, France; [2]Univ. Lille, Inserm, U1003 - PHYCEL - Physiologie Cellulaire, Lille, France; [3]Univ. Lille, Inserm, CHU Lille, Institut Pasteur de Lille, U1011 - EGID, Lille, France; [4]Normandie Université, UNICAEN, INSERM U1086 ANTICIPE, UNICANCER, Cancer Centre F. Baclesse, Caen, France; [5]Univ. Lille, CNRS, UMR8523 - PhLAM – laboratoire de Physique des Lasers, Atomes et Molécules, Lille, France

**Abstract** Control of cell death/survival balance is an important feature to maintain tissue homeostasis. Dependence receptors are able to induce either survival or cell death in presence or absence of their ligand, respectively. However, their precise mechanism of action and their physiological importance are still elusive for most of them including the MET receptor. We evidence that pro-apoptotic fragment generated by caspase cleavage of MET localizes to the mitochondria-associated membrane region. This fragment triggers a calcium transfer from endoplasmic reticulum to mitochondria, which is instrumental for the apoptotic action of the receptor. Knock-in mice bearing a mutation of MET caspase cleavage site highlighted that p40MET production is important for FAS-driven hepatocyte apoptosis, and demonstrate that MET acts as a dependence receptor in vivo. Our data shed light on new signaling mechanisms for dependence receptors' control of cell survival/death balance, which may offer new clues for the pathophysiology of epithelial structures.

*For correspondence:
alessandro.furlan@univ-lille.fr (AF);
david.tulasne@ibl.cnrs.fr (DT)

†These authors contributed equally to this work

Competing interests: The authors declare that no competing interests exist.

## Introduction

The Hepatocyte Growth Factor-Scatter Factor (HGF/SF) receptor MET was discovered three decades ago. Its importance was highlighted during embryogenesis and injury repair, and also in neurodegenerative diseases and cancers (*Furlan et al., 2014*). MET activation triggers a wide variety of biological responses such as survival, proliferation, and migration. The HGF/SF-MET pair notably plays a crucial role in the liver, both during development (*Maina et al., 2001*) and, in adults, in tissue homeostasis and regeneration (*Borowiak et al., 2004*).

In addition to the survival role of ligand-activated MET, the receptor can also promote apoptosis. This has led to classifying it as a dependence receptor. MET cleavage by caspases leads to production of p40MET, an intracellular 40 kDa fragment that can amplify apoptosis, detected in apoptotic primary hepatocytes and mouse livers (*Lefebvre et al., 2013*).

Caspase cleavage of MET occurs at the C-terminal $DNID_{1374}$ and the juxtamembrane $ESVD_{1000}$ sites (mouse sequence). The juxtamembrane site overlaps with the $DY_{1001}R$ site containing a phosphorylated tyrosine ($ESVD_{1000}p$-$Y_{1001}R$) responsible for the recruitment of CBL involved in the receptor degradation (*Peschard et al., 2004*). Although C-terminal caspase cleavage of MET removes only five amino acids in mice, a p40MET fragment to which these C-terminal amino acids remain attached loses its apoptotic potential (*Foveau et al., 2007*; *Ma et al., 2014*).

The p40MET fragment amplifies apoptosis through mitochondrial membrane permeabilization leading to the release of pro-apoptotic factors. Mitochondrial permeabilization is regulated by anti-apoptotic BCL2-family proteins and by pro-apoptotic BH3-only proteins such as BAX and BAK (*Wei et al., 2001*). While specific silencing of BAK significantly inhibits the apoptotic capacity of p40MET, no evidence of direct interaction between p40MET and a BH3-only protein has been reported to date nor the mechanism involved (*Lefebvre et al., 2013*).

Besides their outer mitochondrial membrane (OMM) localization, the pro-apoptotic BH3-only proteins have also been described at the endoplasmic reticulum (ER) (*Zong et al., 2003*). In this organelle, they can interact with calcium channels such as inositol triphosphate receptors (IP3R) and thereby cause deregulation of the calcium flux between the ER and the mitochondria (*Nutt et al., 2002*; *Scorrano et al., 2003*; *Zong et al., 2003*; *Oakes et al., 2005*). Several studies have shown calcium overload in the mitochondrial matrix to cause membrane permeabilization though mitochondrial Permeability Transition Pore (mPTP) opening that can trigger mitochondrial swelling and OMM disruption (for review see *Brenner and Grimm, 2006*).

In the present study, we have evidenced p40MET in the mitochondria-associated endoplasmic reticulum membrane (MAM) region and have characterized p40MET-triggered calcium transfer from the ER to the mitochondria, which is important for its proapoptotic activity. We have also engineered a knock-in mouse model expressing MET mutated at caspase site in order to assess its importance in physiological apoptosis in vivo.

## Results

### p40MET localizes to the interface between the ER and the mitochondria

To gain insights into the pro-apoptotic function of MET, we compared p40MET fragment with its non-apoptotic variant p40MET D1374N, mutated at the C-terminal caspase site, both fragments fused to Enhanced Green Fluorescent Protein (GFP) (*Figure 1a*; *Figure 1—figure supplement 1a*). In MCF10A epithelial cells, ectopic expression of GFP-p40MET induced cytochrome C release and caspase 3 cleavage as efficiently as the previously published flag-p40MET, in contrast to GFP-p40MET D1374N or GFP alone (*Figure 1b–c*; *Figure 1—source data 1*; and representative pictures in *Figure 1—figure supplement 1b–c*). Furthermore, in p40MET-transfected cells, treatment with a pan-caspase inhibitor increased the proportion of cells displaying cytochrome-C release, a likely consequence of an inhibition of late apoptosis, which would impair apoptotic cell detachment and favor their detection (*Figure 1—figure supplement 1d*; *Figure 1—source data 1*). To assess the proapoptotic property of MET in hepatocytes, similar experiments were performed in immortalized human hepatocytes (IHH) (*Schippers et al., 1997*). In these cells, apoptosis induced by BH3 mimetic promoted p40MET generation, which was inhibited by pan-caspase inhibitor treatments and by MET silencing (*Figure 1—figure supplement 2*). Moreover, ectopic expression of GFP-p40MET in this cell line induced caspase 3 cleavage (*Figure 1d*; *Figure 1—source data 1*).

Because of a partial overlap between p40MET and mitochondrial signals acquired by immunofluorescence (*Lefebvre et al., 2013; Figure 1—figure supplement 3*), we wondered whether p40MET might localize to a peculiar zone very close to mitochondria, namely the MAM region. MAMs constitute a subdomain with direct interactions between the ER and mitochondria. They notably play a crucial role in amplifying cell death (*Naon and Scorrano, 2014*). We thus examined whether p40MET might co-localize with Fatty Acid CoA Ligase 4 (FACL4), a protein residing mostly in MAMs, and this proved to be the case (*Figure 1e*; *Figure 1—source data 1*). FACL4 was also found to colocalize with the p40MET D1374N fragment, but not with GFP. To confirm this localization via another approach, endogenous p40MET was generated by an apoptotic stress and subcellular fractionation was performed. Mitochondrial fraction was enriched in the mitochondrial $Ca^{2+}$ uniporter (MCU) and an ER fraction in the ER chaperon protein calnexin. The MAM fraction displayed both MCU and calnexin, as expected, and was enriched in FACL4 (*Figure 1f*). p40MET was found exclusively in the MAM fraction, in agreement with the immunofluorescence.

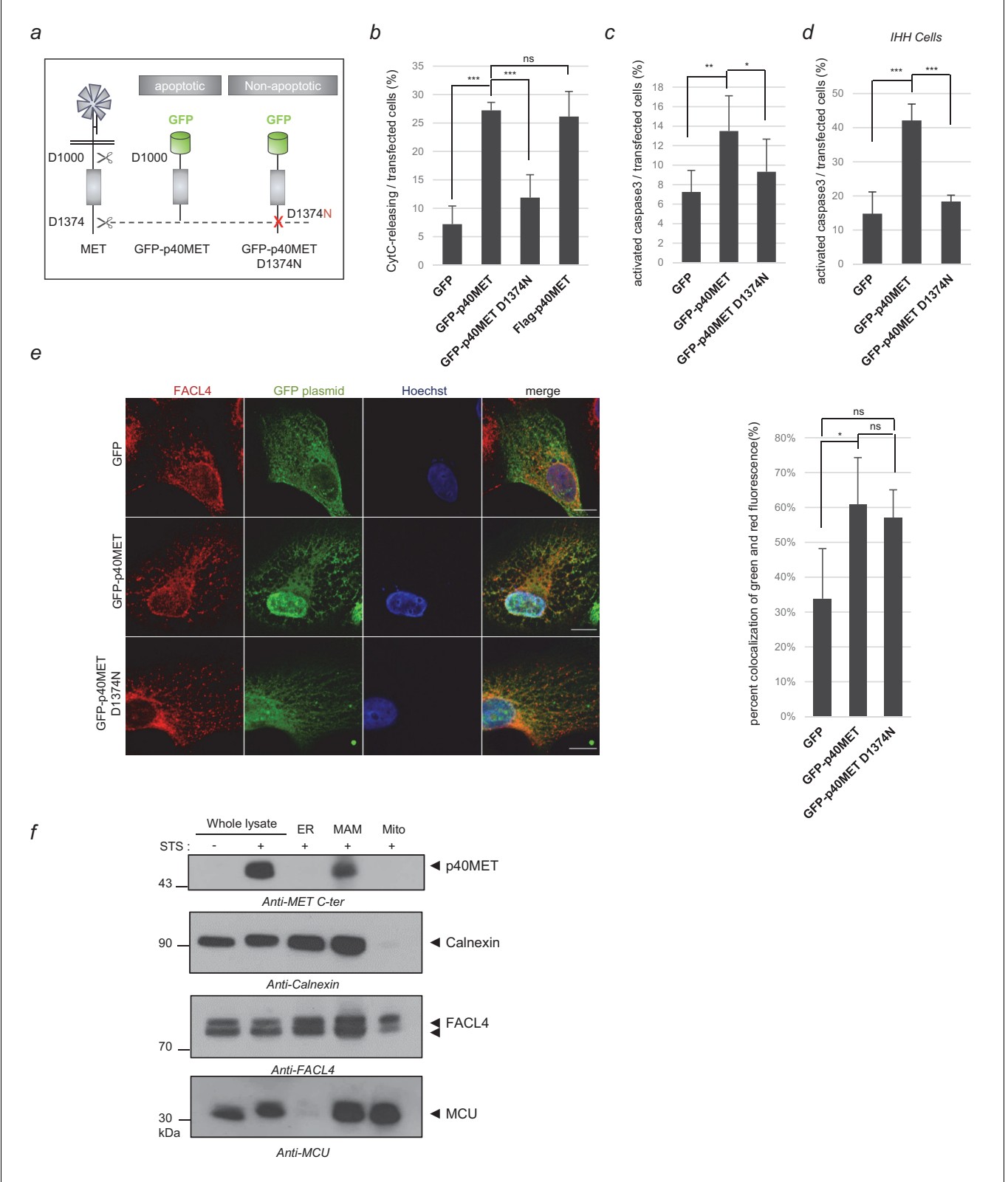

**Figure 1.** Locating the p40MET fragment by immunofluorescence and subcellular fractionation. (a) Schematic representation of MET receptor cleavage by caspase during apoptosis, with besides representation of the GFP-p40MET fragment and of the GFP-p40MET D1374N fragment which still possesses the C-terminal tail. (b, c, d) MCF10A (b–c) or IHH (d) cells were transiently transfected with a vector expressing GFP, GFP-p40MET, GFP-p40MET D1374N, or flag-p40MET. After 24 hr transfection for MCF10A and 48 hr for IHH, the cells were fixed and labeled with an appropriate

*Figure 1 continued on next page*

*Figure 1 continued*

antibody: anti-flag when transfected with the vector expressing flag-p40MET, anti-cytochrome C or anti-cleaved caspase 3 antibody to evaluate apoptosis. The percentage of cytochrome C release or of cleaved-caspase-3-positive cells was determined with respect to the number of GFP- or flag-positive cells. At least 150 cells per well ($n = 6; \pm$ S.D.) (b), 200 cells per well ($n = 6; \pm$ S.D.) (c) and 60 cells per well ($n = 4; \pm$ S.D.) (d) were counted. (e) MCF10A epithelial cells were transfected with a vector expressing GFP, GFP-p40MET, or GFP-p40MET D1374N. Twenty-four hours after transfection, the nuclei were stained with Hoechst (blue staining) and immunofluorescence staining was performed with anti-FACL4 to label the MAMs (red staining). Cells were observed by fluorescence confocal microscopy. Weighted colocalization coefficients were determined by means of Manders coefficients for green staining (of GFP, GFP-p40MET, or GFP-p40MET D1374N) and FACL4 staining (red) on the basis of the fluorescence confocal microscopy images ($n = 30; \pm$ S.D.). (f) MCF10A cells were starved overnight and treated for 4 hr with 1 µM staurosporine (STS). After treatment, the cells were fractionated into ER, MAM and mitochondrial fractions. Proteins from whole-cell lysates (50 µg) and from the different fractions (20 µg) were analyzed by western blotting with antibodies against the MET kinase domain, the reticular protein calnexin, the MAM protein FACL4, and the inner mitochondrial membrane protein MCU. The positions of prestained molecular weight markers are indicated. Arrows indicate the positions of p40MET, calnexin, FACL4 and MCU; scale bar = 10 µm, ns, nonsignificant; *, $p<0.05$; **, $p<0.01$; ***, $p<0.001$ as determined by Student's *t* test.

The online version of this article includes the following source data and figure supplement(s) for figure 1:

**Source data 1.** Source data of *Figure 1b–c–d* and *Figure 1—figure supplement 1d* reporting counting of GFP, active caspase 3, and cytochrome C release positive cells, calculation of the percentage, mean and SD, diagram conception and statistical analyses; source data of *Figure 1e* reporting the coefficient of fluorescence colocalisation, calculation of the mean, SD and statistical analyses.

**Figure supplement 1.** Validation of the vectors expressing GFP-p40MET and GFP-p40MET D1374N.

**Figure supplement 2.** p40MET fragment generation in IHH cells.

**Figure supplement 3.** Partial overlap between GFP-p40MET and mitotracker signals acquired by immunofluorescence.

## The p40MET fragment partners with the BH3-only protein BAK

The subcellular localization of the proapoptotic p40MET fragment prompted us to further investigate functional relationship between p40MET and BCL2 family proteins. Indeed, beside their localization at the outer membrane of mitochondria, BCL2 proteins were found localized at the ER, both localization being involved in apoptosis regulation (*Scorrano et al., 2003*; *Rong et al., 2008*). To determine the putative involvement of the pro-apoptotic BCL2 family proteins in p40MET-induced apoptosis, BAK, BAX or BOK were silenced in p40MET transfected MCF10A (*Figure 2—figure supplement 1a*) and IHH cells (*Figure 2—figure supplement 2a*). In MCF10A, BAK siRNA abrogated MET fragment induced caspase 3 activation and cytochrome C release, BOK silencing partially inhibited them, while BAX silencing had no effect (*Figure 2—figure supplement 1b–c*; *Figure 2—figure supplement 1—source data 1*). Similar results were obtained in IHH cells regarding p40MET induced caspase 3 activation (*Figure 2—figure supplement 2b*; *Figure 2—figure supplement 2—source data 1*). These results are consistent with our previous study on the impact of BAK and BAX silencing on the apoptosis of MCF10A cells transfected with p40MET-Flag (*Lefebvre et al., 2013*) and confirmed the main involvement of BAK in p40MET-induced apoptosis. To elucidate whether p40MET actually interacts with BAK and other BCL2 family members, molecular engineering was carried out to fuse these proteins with mTurquoise2 (mT2) or Super Yellow Fluorescent Protein 2 (SYFP2) in order to perform FRET (Förster Resonance Energy Transfer) experiments (*Figure 2a*; *Figure 2—figure supplement 3a–b*). Previous experiments had determined that this fluorophore couple was convenient for FRET quantification by fluorescence lifetime (*Bidaux et al., 2018*). The FRET lifetime analysis is based on the fact that transfer energy between the donor (mT2 here) and its acceptor (SYFP2) accelerates the fluorescent decay of the donor. Although maximum FRET efficiency measured in that way hardly reaches 10%, it is interestingly not dependent on the fluorophore concentration in contrast to the intensity-based method. As a positive control, BAK-mT2 and BCL-XL-SYFP2 co-transfection led to a mean FRET efficiency of 8% (in agreement with what can be obtained for other heteromeric proteins), while it was negligible (<0.1%) with vector encoding freely diffusing SYFP2 (*Figure 2b*; *Figure 2—source data 1*). Co-transfection with the vectors encoding BAK-mT2 and p40MET-SYFP2also gave rise to FRET about 2.5% efficiency (*Figure 2b*; *Figure 2—source data 1*). In the reverse donor-acceptor configuration, we found a mean FRET efficiency of 4% confirming the interaction (*Figure 2c*; *Figure 2—source data 1*). We also noticed a FRET phenomenon occurring between p40MET and BAX (with mean values respectively at 2.5% and 4.3%) (*Figure 2—figure supplement 4a–b*; *Figure 2—figure supplement 4—source data 1*). In contrast, p40MET-mT2 hardly interacted with BCL-XL (1.5% mean FRET efficiency). These results are to be compared with the 1% FRET efficiency recorded for negative control cells co-transfected with freely diffusing SYFP2

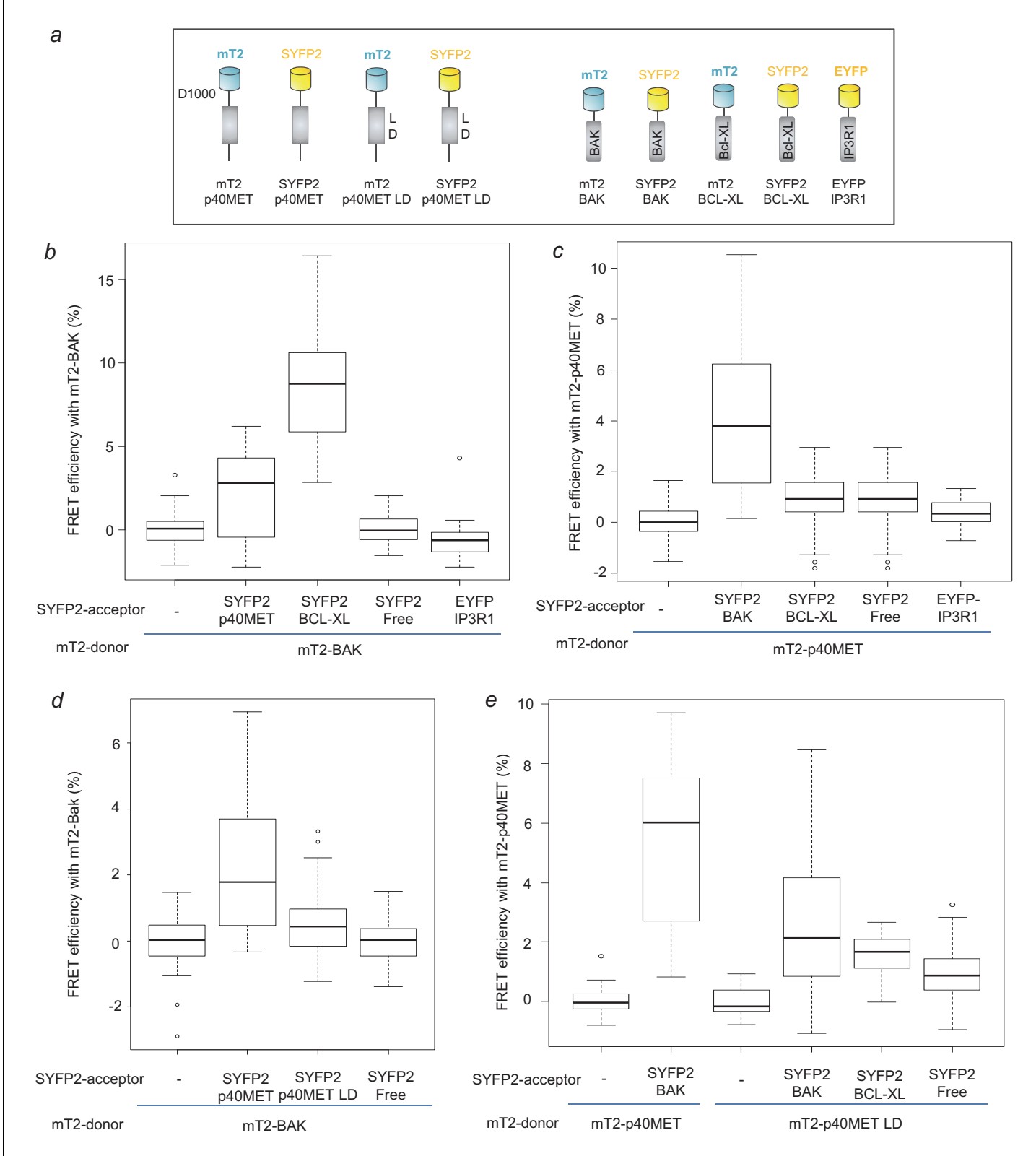

**Figure 2.** FRET measurement to analyze p40MET and BAK interaction. (**a**) Schematic representation of the mTurquoise2- or SYFP2-tagged proteins used for FRET analysis. (**b**) Cells were co-transfected with a vector expressing BAK-mT2 as FRET donor and with a vector expressing free SYFP2, BCL-XL-SYFP2, EYFP-IP3R1 or p40MET-SYFP2 as FRET acceptor. The fluorescence lifetime of BAK-mT2 was measured by Time-Correlated Single Photon Counting (TCSPC) and the FRET efficiency was calculated with respect to the donor-alone condition. At least 35 cells were counted for each condition.
*Figure 2 continued on next page*

*Figure 2 continued*

(c) Cells were co-transfected with a vector expressing p40MET-mT2 as FRET donor and a vector expressing free SYFP2, BAK-SYFP2, EYFP-IP3R1 or BCL-XL-SYFP2 as FRET acceptor. The fluorescence lifetime of p40MET-mT2 was measured by TCSPC and the FRET efficiency was calculated with respect to the donor-alone condition. At least 40 cells were counted for each condition. (d and e) Cells were co-transfected with a vector expressing BAK-mT2 as FRET donor and with a vector expressing free SYFP2, p40MET-SYFP2 mutated or not at the L and D residues of the putative BH3-like domain as FRET acceptor (d). Inversely, cells were co-transfected also with vectors expressing p40MET-mT2 mutated or not on the L and D residues as FRET donor and a vector expressing free SYFP2, BAK-SYFP2, or BCL-XL-SYFP2 as FRET acceptor (e). The fluorescence lifetime of p40MET-mT2 and BAK-mT2 were measured by TCSPC and the FRET efficiency was calculated with respect to the donor-alone condition. At least 30 cells were counted for each condition.

The online version of this article includes the following source data and figure supplement(s) for figure 2:

**Source data 1.** Source data of *Figure 2b-e* reporting FRET source data and diagram conception.
**Figure supplement 1.** Effect of BAK, BAX and BOK silencing on p40MET-induced apoptosis in MCF10A cells.
**Figure supplement 1—source data 1.** Source data of *Figure 2—figure supplement 1b–c* reporting counting of GFP, active caspase 3, and cytochrome C release positive cells, calculation of the percentage, mean and SD, diagram conception and statistical analyses.
**Figure supplement 2.** Effect of BAK, BAX and BOK silencing on p40MET-induced apoptosis in IHH cells.
**Figure supplement 2—source data 1.** Source data of *Figure 2—figure supplement 2b* reporting counting of GFP and active caspase 3 positive cells, calculation of the percentage, mean and SD, diagram conception and statistical analyses.
**Figure supplement 3.** Validation of the vectors expressing m-Turquoise2- or SYFP2-tagged proteins in FRET experiments.
**Figure supplement 4.** FRET experiments between p40MET and BCL2 family members.
**Figure supplement 4—source data 1.** Source data of *Figure 2—figure supplement 4a–e* including FRET source data and diagram conception.
**Figure supplement 5.** Consequence of putative BH3 domain mutation on p40MET-induced apoptosis.
**Figure supplement 5—source data 1.** Source data of *Figure 2—figure supplement 5b–c* reporting counting of GFP, active caspase 3 and cytochrome C release positive cells, computation of the percentage, mean and SD, diagram conception and statistical analyses.

(*Figure 2c*; *Figure 2—source data 1*). In a similar way, the study of p40MET interaction with BOK resulted in negative output (*Figure 2—figure supplement 4a–c*; *Figure 2—figure supplement 4—source data 1*) whereas a strong interaction between BOK and BCL-XL was evidenced.

In order to confirm the importance of p40MET interaction with BAK in hepatic cells, we carried out FRET experiments with these proteins in IHH cells. We actually found a 1.5% FRET efficiency (p<0.05) for a transfer from mT2-p40MET to SYFP2-BAK and 5.0% for a transfer from mT2-BAK to SYFP2-p40MET (p<0.001) (*Figure 2—figure supplement 4d–e*; *Figure 2—figure supplement 4—source data 1*). The interaction between p40MET and BAK is thus confirmed in these liver-derived cell lines.

Since BAK can control apoptosis via the modulation of ER $Ca^{2+}$ release, we wondered whether we might detect interactions between BAK or p40MET and IP3Rs. No FRET was detected between BAK-mT2 or p40MET-mT2 and IP3R1-EYFP (*Figure 2b–c*; *Figure 2—source data 1*). Hence, this last part of experiment does not allow us to conclude on the putative involvement of IP3R in p40MET-triggered apoptosis. Interactions between BCL2 family members involve the BH3 (BCL2 homology 3) domain found notably in BH3-only protein such as BID or BIM but also in BH3-like proteins including the receptor tyrosine kinase ERBB2 and ERBB4 (*Naresh et al., 2006*; *Strohecker et al., 2008*). Alignment of BH3 domain from BH3-only and BH3-like proteins with p40MET sequence revealed a putative BH3 domain with an LXXXXD core motif conserved in mice and humans (*Figure 2—figure supplement 5a*). Mutation of the corresponding L1110 and D1115 residues in GFP-p40MET abrogated its pro-apoptotic activity (*Figure 2—figure supplement 5b–c*; *Figure 2—figure supplement 5—source data 1*). Furthermore, FRET experiments with plasmids encoding mT2-p40MET and SYFP2-p40MET mutated or not on L and D residues (*Figure 2a*; *Figure 2—figure supplement 3a–b*) showed that mutation of the potential BH3-like domain of p40MET decreased its interaction with BAK (*Figure 2d–e*; *Figure 2—source data 1*). Taken together, these results suggest that p40MET interacts directly with BAK through a putative BH3-like domain and that this partner might participate in the pro-apoptotic action of p40MET.

## Apoptosis amplification by p40MET involves calcium flux deregulation

Under stress conditions, the ER can release $Ca^{2+}$ via functional units present mostly in MAMs. Within MAMs, IP3R channels are very close to the MCU channels of the mitochondria, and this allows $Ca^{2+}$ uptake by the mitochondria. $Ca^{2+}$ overload in the mitochondria leads to their permeabilization and

ultimately to apoptosis (*Naon and Scorrano, 2014*). Because, BAK has been described to play a role in this process by promoting $Ca^{2+}$ release from the ER (*Scorrano et al., 2003*), we checked for a possible involvement of $Ca^{2+}$ in p40MET-induced apoptosis. We found p40MET-induced cytochrome C release to be inhibited in MCF10A cells cultured in $Ca^{2+}$-free medium (*Figure 3a*; *Figure 3—source data 1*). In addition, $Ca^{2+}$ chelation prevented p40MET-induced cytochrome C release in MCF10A cells or caspase 3 activation in IHH (*Figure 3b–c*; *Figure 3—source data 1*). Similar inhibition of cytochrome C release in MCF10A were observed upon treatment with xestospongin B, a potent IP3R inhibitor (*Figure 3d*; *Figure 3—source data 1*), or with an siRNA against MCU (*Figure 3e–f*; *Figure 3—source data 1*). Apoptosis induced by staurosporine was unaffected by such treatments (*Figure 3—figure supplement 1a–c*). Taken together, our data demonstrate that a $Ca^{2+}$ flux between the ER and mitochondria is required for p40MET-induced cytochrome C release from the mitochondria.

To further evaluate the impact of p40MET on $Ca^{2+}$ exchange, we used thapsigargin, a sarcoplasmic-endoplasmic reticulum $Ca^{2+}$ ATPase (SERCA) inhibitor. SERCA inhibition causes the depletion of the ER $Ca^{2+}$, allowing indirect measurement of the ER $Ca^{2+}$ concentration with Fura-2, a cytosolic calcium probe. After thapsigargin addition, GFP-p40MET-expressing cells showed a significantly lower Fura-2 fluorescence peak than cells expressing GFP-p40MET D1374N or GFP (*Figure 3g*). This suggests that the ER $Ca^{2+}$ pool was depleted after p40MET transfection. To see if this depletion was accompanied by $Ca^{2+}$ accumulation in the mitochondria, we used a mitochondrion-specific biosensor, 4mtD3cpv. This probe consists of a fluorescent FRET couple linked to a calmodulin binding site, allowing $Ca^{2+}$-dependent modulation of FRET. We found p40MET transfection to increase the FRET efficiency by 26% (p=0.0013) as compared to transfection with a control plasmid, suggesting a substantial increase of the calcium concentration in mitochondria (*Figure 3h*; *Figure 3—source data 1*). Lastly, we examined whether mPTP pore opening, which can be driven by mitochondrial calcium overload, might participate in p40MET-induced apoptosis. As suspected, mPTP inhibition by cyclosporin A was found to reduce the p40MET-induced apoptosis (*Figure 3i*; *Figure 3—source data 1*). Altogether, these data show that p40MET impairs the ER-to-mitochondria calcium homeostasis, eventually causing a mitochondrial calcium overload and permeabilization.

## Generation of a new MET knock-in mouse model and derived primary cell lines to study MET pro-apoptotic activity

To investigate the in vivo involvement of MET cleavage in apoptosis, we used Cre-Lox recombination to develop a knock-in mouse model where the *Met* gene locus is modified in the C-terminal caspase site (*Figure 4a–b*). The mutation introduced is D1374N, as in the above-described in vitro experiments, and it prevents C-terminal cleavage. We chose to mutate residue D1374 rather than the juxtamembrane residue D1000 because the latter also belongs to the DYR sequence driving Cbl recruitment, mutating it would probably alter both mechanisms.

As in vivo studies on apoptosis classically induce the animal's death and in order to comply with animal model ethics recommendations, we chose first to derive primary cell models to validate the relevance of MET cleavage by caspases ex vivo and to evaluate the response to several apoptotic conditions. We focused on primary cells that are long-lived and do not require animal dissection for each experiment. First results showing D1374N Mouse Embryonic Fibroblasts (MEFs) to be more resistant than wild-type to anisomycin treatment evidenced the importance of the caspase cleavage site for optimal apoptosis (*Figure 4—figure supplement 1a–b*; *Figure 4—figure supplement 1—source data 1*). MEFs, however, produce relatively low levels of MET receptor (*Foveau et al., 2007*), so we chose to derive more relevant long-lived cells, namely Bipotential Mouse Embryonic Liver (BMEL) cells to analyze MET involvement in the liver survival-apoptosis balance. Several WT and MET D1374N BMEL clones were derived from embryonic livers. We selected two WT clones (A and B) and two D1374N clones (A and B) displaying comparable BMEL marker expression quantified by RT-Q-PCR (*Krt19*, *Hnf4a* as BMEL markers; *Aldob*, *Alb,* as markers of differentiated hepatocytes) (*Figure 4c*; *Figure 4—figure supplement 1c*). We first found WT and D1374N BMEL clones to show comparable doubling times (*Figure 4d*). In addition, HGF/SF stimulation induced islet scattering with equal efficiency in the WT and D1374N clones (*Figure 4e*). The D1374N MET mutation likewise did not alter either HGF-induced MET phosphorylation or AKT and ERK phosphorylation (*Figure 4f*). Similar results were obtained with the other pair of WT and D1374N BMEL clones (*Figure 4—figure supplement 1d–e*). Lastly, massive apoptosis induced by a 7 hr staurosporine treatment led to

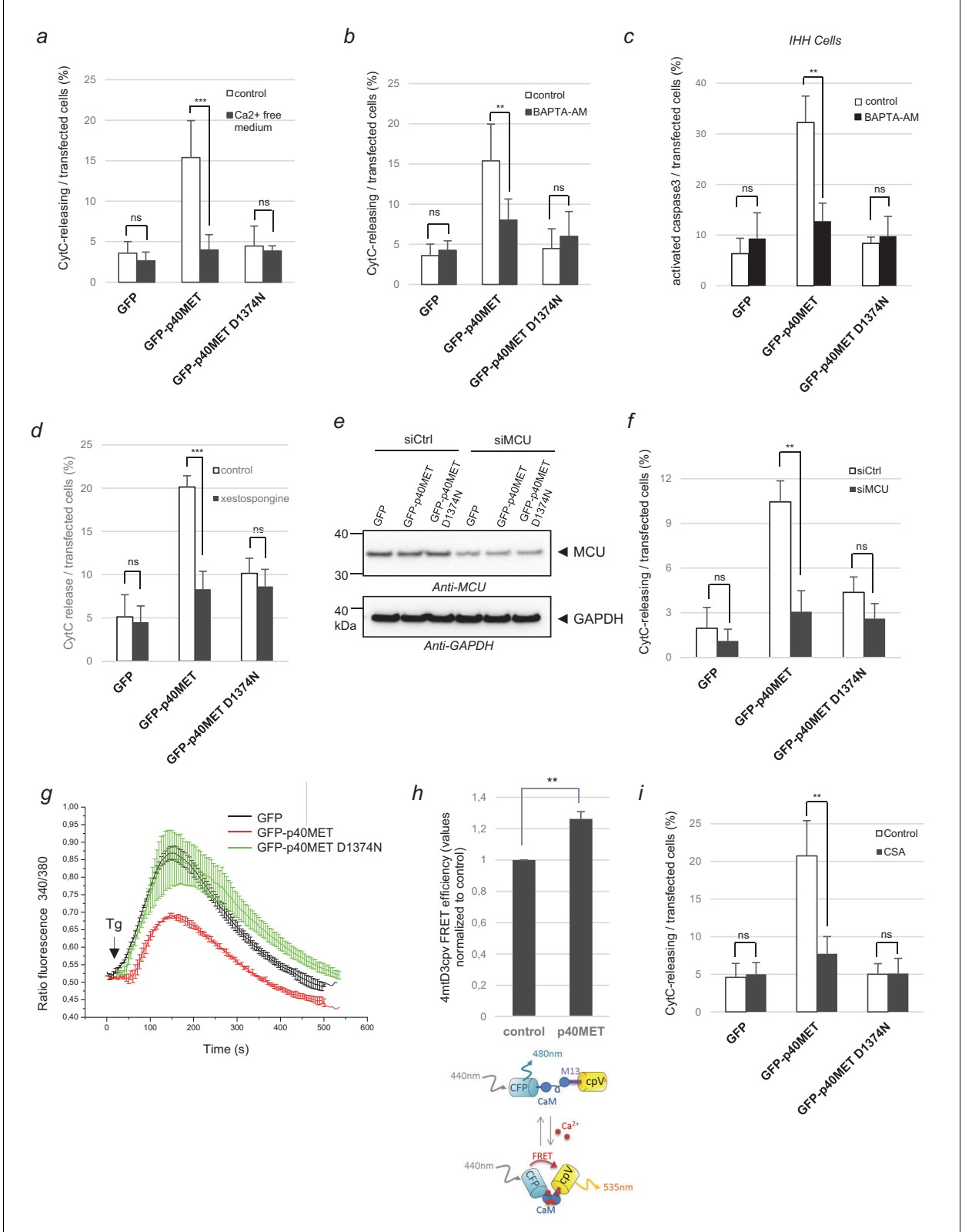

**Figure 3.** Evaluation of p40MET-induced apoptosis after inhibition of calcium exchanges between the ER and mitochondria. (a, b) MCF10A epithelial cells were starved overnight in a calcium-free medium or treated with the calcium chelator BAPTA-AM (10 µM). The next day cells were transfected with a vector expressing a GFP-tagged fragment. (c) IHH cells were transfected with a vector expressing a GFP-tagged fragment and treated the next day with the calcium chelator BAPTA-AM (10 µM) for 24 hr. (d) MCF10A cells were transiently transfected with a vector expressing GFP, GFP-p40MET, or

*Figure 3 continued on next page*

*Figure 3 continued*

GFP-p40MET D1374N and treated or not with the IP3R inhibitor xestospongin-B (5 μM). (a–d) After 24 hr transfection for MCF10A and 48 hr for IHH, cells were fixed and processed for immunostaining with an anti-cytochrome C or anti-cleaved caspase 3 antibody to evaluate apoptosis. The percentage of cytochrome C release or of cleaved caspase-3-positive cells was determined with respect to the number of GFP positive cells. At least 100 cells were counted per well (n = 6;± S.D.) for MCF10A and at least 60 cells (n = 4;± S.D.) for IHH. (e–f) One day before transfection with a vector expressing a GFP-tagged fragment, MCF10A cells were transfected with a control siRNA or with a mixture of three siRNAs targeting the mitochondrial calcium channel MCU. Twenty-four hours later, (e) one part of the cells was lysed and extracts were analyzed by western blotting with an anti-MCU and an anti-GAPDH antibody. (f) For immunofluorescence, staining was performed with an anti-cytochrome-c antibody and nuclei were labeled with Hoechst. The percentage of transfected cells displaying cytochrome-c release was determined. At least 60 cells were counted per well (n = 3;± S.D.). (g) HEK293 cells were transfected with a vector expressing a GFP-tagged fragment. The next day, the cells were incubated in $Ca^{2+}$-free HBS solution and treated with 1 μM Thapsigargin (Tg). The calcium concentration was determined by estimating the uncorrected 340 nm/380 nm fluorescence ratio of fura-2AM. At least 20 cells were measured per condition (n = 3;± S.D.). The presented results are representative of three independent experiments. Black arrows indicate Tg injection. (h) HEK293 cells were co-transfected with the 4mtD3cpv biosensor and a plasmid encoding either GFP or GFP-p40MET. The CFP fluorescence lifetime was recorded and the FRET efficiency, indicative of the calcium level in the mitochondria, was calculated with respect to the level observed for the control, set as reference. At least 30 cells were counted for each condition (n = 3;± S.D.). Below, schematic representation of the 4mtD3cpv biosensor constituted by two fluorescent probes linked by a calmodulin binding site, allowing FRET measurement. (i) MCF-10A epithelial cells were transiently transfected with a vector expressing GFP, GFP-p40MET, or GFP-p40MET D1374N and were treated or not with 2.5 μM cyclosporinA (CSA). Twenty-four hours after transfection, the nuclei were stained with Hoechst and immunofluorescence staining was performed with an anti-Flag antibody and an anti-cytochrome C antibody. The percentage of MET-transfected cells displaying cytochrome C release was determined. At least 200 cells were counted per well (n = 3;± S.D.). ns, non significant; *, p<0.05; **, p<0.01 as determined by Student's *t* test. The online version of this article includes the following source data and figure supplement(s) for figure 3:

**Source data 1.** Source data of *Figure 3a–b–c–d–f–i* reporting counting of GFP, active caspase 3 and cytochrome C release positive cells, computation of the percentage, mean and SD, diagram conception and statistical analyses; source data of *Figure 3h* including FRET source data and diagram conception.

**Figure supplement 1.** Involvement of calcium flux in staurosporine-induced apoptosis.

---

generation of the p40MET fragment in the WT cells and of a slightly longer p40MET D1374N fragment in the D1374N cells, in agreement with the presence of the C-terminal tail, which was prevented by pan-caspase inhibitors (*Figure 4g*). In conclusion, hepatic progenitors from WT and D1374N mice responded similarly to HGF/SF but produced, under apoptotic conditions, either p40MET or the longer p40MET D1374N fragment.

## Hepatic progenitors depend on p40MET to complete the apoptosis process

A first hint regarding the relative sensitivities of WT and MET D1374N cells to apoptosis was obtained by removing the collagen coating and comparing the abilities of these cells to adapt to this stress. Whereas the WT cells failed to grow under these conditions, MET D1374N BMEL cells formed small clusters from day 2, showing a better capacity to cope with the loss of matrix-driven signals (*Figure 5a*; *Figure 5—figure supplement 1*). Next, a 4 hr treatment with staurosporine, used to induce partial apoptosis, proved suitable for comparing cell death in the two cell lines. While staurosporine treatment induced caspase 3 and PARP cleavages in the WT cells, these cleavages were undetected in the MET D1374N cells (*Figure 5b*). Immunostaining for cleaved caspase 3 further showed the MET D1374N cells to be more resistant than the WT cells to staurosporine-induced apoptosis (*Figure 5c*; *Figure 5—source data 1*). Resistance to apoptosis in MET D1374N cells was also observed upon BH3 mimetic ABT737 treatment (*Figure 5—figure supplement 2*). These results demonstrate the importance of MET cleavage for apoptosis amplification in hepatic progenitors.

To assess how calcium flux affects apoptosis in this cell model, we evaluated caspase 3 activation after $Ca^{2+}$ flux inhibition. First, a lower percentage of MET D1374N BMEL cells than WT cells displayed staurosporine-induced caspase activation as shown above (*Figure 5d*; *Figure 5—source data 1*). Second, only the WT population responded to the additional presence of the IP3R inhibitor xestospongin, confirming involvement of $Ca^{2+}$ flux for p40MET induced cell death. Next, to evaluate mitochondrial $Ca^{2+}$ uptake by WT and MET D1374N BMEL cells, we assessed the capacity of BMEL-cell mitochondria to buffer cytoplasmic $Ca^{2+}$ increases upon addition of exogenous $Ca^{2+}$. We treated WT and MET D1374N BMEL cells with staurosporine, permeabilized and incubated them with a calcium-green fluorescent probe, and injected exogenous $Ca^{2+}$. A rapid and transient increase in cytosolic $Ca^{2+}$ was followed by a decrease due to uptake of the $Ca^{2+}$ excess by the mitochondria

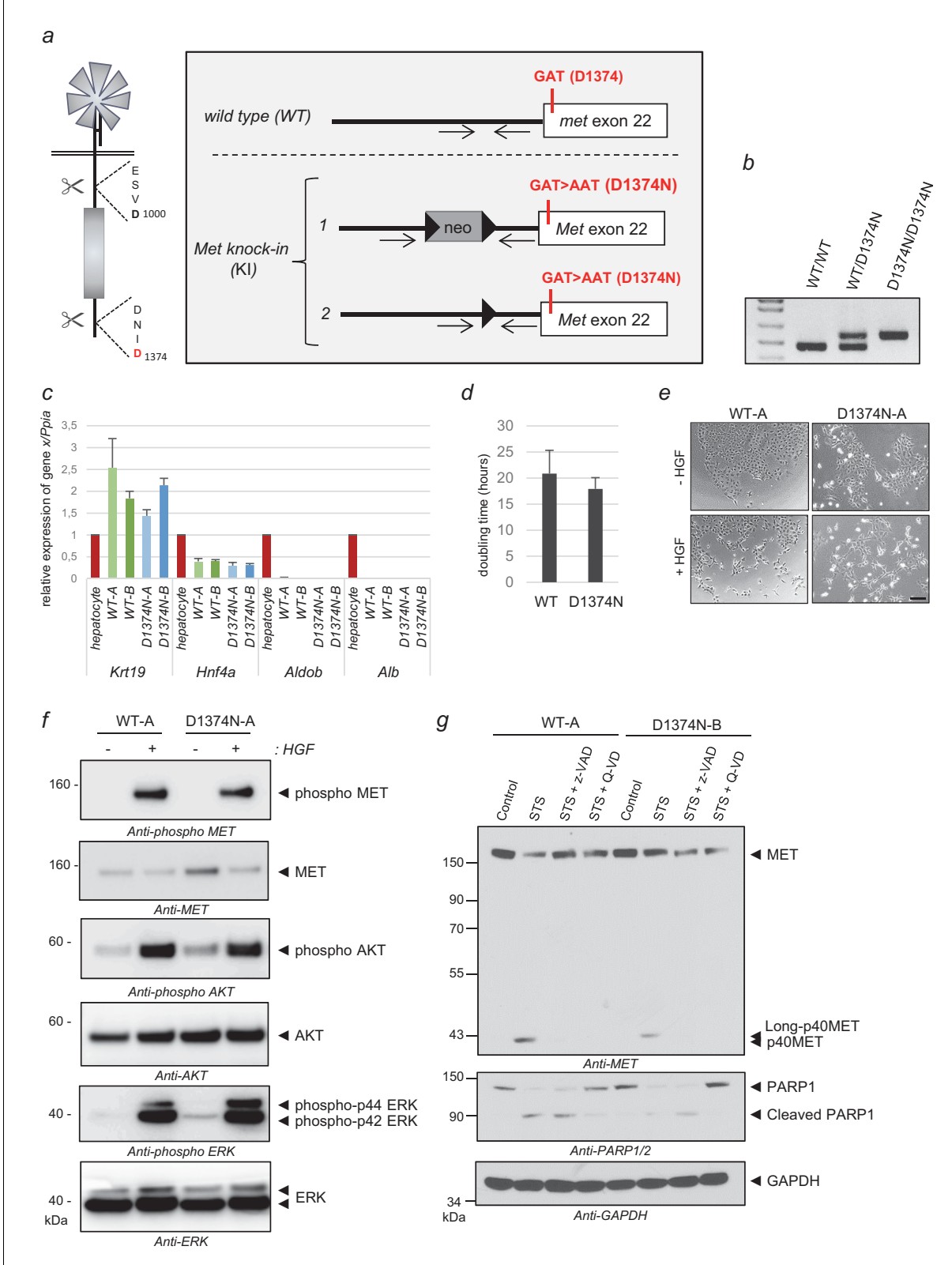

**Figure 4.** Generation of MET knock-in mice mutated at the C-terminal caspase cleavage site and isolation of hepatic progenitors. (**a**) Strategy for generating knock-in mice. The targeted *Met* allele is depicted. White boxes represent *Met* exon 22 with, in red, the generated knock-in (KI) mutation GAT >AAT (D1374N). Bold black arrowheads indicate LoxP sites. The positions of the genotyping primers are marked with thin black arrows. (**b**) To confirm the presence of the KI mutation in *Met* exon 22, PCR genotyping was performed with primers flanking the loxP sites and amplifying a 476 bp of

*Figure 4 continued on next page*

*Figure 4 continued*

the wild-type allele (WT) and a 563 bp fragment of the *Met* KI allele (D1374N). (**c**) Levels of *Krt19*, *Hnf4a*, *Aldob*, and *Alb* transcripts were measured by RT-qPCR in Bipotential Mouse Embryonic Liver cell (BMEL) clones derived from WT and MET D1374N mice, and murine hepatocytes as a control. Analyses of two WT BMEL clones (Clones A and B) and two D1374N clones (clones A and B) are shown. The results presented are averages of three independent experiments, with errors bars showing standard deviations. (**d**) BMEL cells were cultured under routine conditions and were counted after 24, 48, or 72 hr and the doubling times of the WT and MET D1374N BMEL cells were established by averaging the values obtained for the two corresponding clones. (**e**) Cells were seeded at low confluence. The next day the cells were starved for 30 min in the presence or absence of 10 ng/ml HGF/SF. Representative pictures were taken after 24 hr; scale bar = 100 μm (**f**) BMEL cells were starved overnight in RPMI-0% FCS and stimulated or not for 10 min with 20 ng/ml HGF/SF. For each condition, the same amount of whole cell lysate was analyzed by western blotting with antibodies against mouse MET, ERK, AKT, and their phosphorylated forms. (**g**) BMEL cells were treated for 7 hr with 1 μM staurosporine (STS) with or without the pan-caspase inhibitor zVAD-FMK or Q-VD (20 μM). The same amount of protein was resolved by SDS-PAGE and analyzed by immunoblotting with antibodies against the MET kinase domain and cleaved PARP, to assess apoptosis induction, and GAPDH, to assess loading.

The online version of this article includes the following source data and figure supplement(s) for figure 4:

**Figure supplement 1.** Culture phenotype of MEF and BMEL cells and responses of WT-B and D1374N-B clones to survival or apoptosis induction.
**Figure supplement 1—source data 1.** Source data of *Figure 4—figure supplement 1b* reporting counting of active caspase 3 positive cells according to the number of cells per field, computation of the mean, percentage, SD, diagram conception and statistical analyses.

(*Figure 5e*). This process was repeated until mitochondrial uptake was no longer possible, as attested by a fluorescence plateau. After the first injections, MET D1374N cells took up $Ca^{2+}$ more efficiently than WT cells. Furthermore, on average, mitochondrial uptake stopped in the WT cells at a total concentration of 100 μM, versus 140 μM for the MET D1374N cells (*Figure 5e–f*; *Figure 5—source data 1*). This lower $Ca^{2+}$ uptake capacity of staurosporine-treated WT BMEL cells as compared to MET D1374N BMEL cells suggests that the former are already overloaded with $Ca^{2+}$, possibly because of p40MET generation.

## C-terminal caspase cleavage of MET is required in vivo for optimal apoptosis in the mouse liver

WT and MET D1374N mice displayed no obvious phenotypic differences and interbreeding between WT, D1374N, and heterozygous mice gave the expected Mendelian distributions (*Figure 6a–b*). D1374N mice did not display any notable developmental anomalies in agreement with a functional pro-survival MET signaling. We also looked more carefully at the organization of the liver and mammary glands, organs known to require MET activity for their proper morphogenesis, which displayed normal organization (*Figure 6c*; *Figure 5—figure supplement 1*).

To investigate the sensitivity of WT and MET D3174N mouse liver tissues to apoptosis, we focused on FAS-induced fulminant hepatitis, a condition in which HGF/SF production is reported to promote hepatocyte survival (*Kosai et al., 1998*). To prevent any survival response induced by ligand-activated MET, mice were treated beforehand with crizotinib, a potent clinically used MET inhibitor (*Zhang et al., 2016*). HGF/SF was indeed able to induce a survival response in primary hepatic progenitors and this response was reduced by crizotinib (*Figure 6—figure supplement 2a–b*) Therefore, mice received crizotinib orally before administration of FAS agonistic antibody by intraperitoneal injection. Caspase 3 activation was scored by the mean of image analysis on liver slices, according to the proportion of stained area (from negative – (0% to 2%) to highly positive +++ (>10%)). As expected, the livers of animals not having received a FAS injection were negative for cleaved caspase 3 staining (*Figure 6d*; *Figure 6—figure supplement 3a*). Four hours after FAS injection, the WT mice displayed more caspase 3 activation than D1374N mice, with respectively 62% (8/13) and 27% (2/15) highly positive livers. Conversely, only one WT liver out of 13 (8%) was found negative for caspase 3 activation, as opposed to 47% (7/15) of the D1374N livers (*Figure 6e*; *Figure 6—source data 1*; *Figure 6—figure supplement 3b* for a complete-panel illustration), clearly demonstrating the resistance of MET D1374N mice to apoptosis induction, with respect to WT mice. Overall, 92% (12/13) of the WT mice were positive for cleaved caspase 3 staining, as compared to 53% (8/15) of the MET D1374N mice. Dosage of plasma transaminase activities, ALAT (alanine aminotransferases) and ASAT (aspartate aminotransferases), can also be used to monitor liver injury. In independent experiments, ALAT and ASAT concentrations were therefore measured in plasma from WT and MET D1374N mice before and after FAS-agonist treatment. The resistance of MET D1374N mice to FAS-induced apoptosis was corroborated by these dosages, since WT mice displayed a

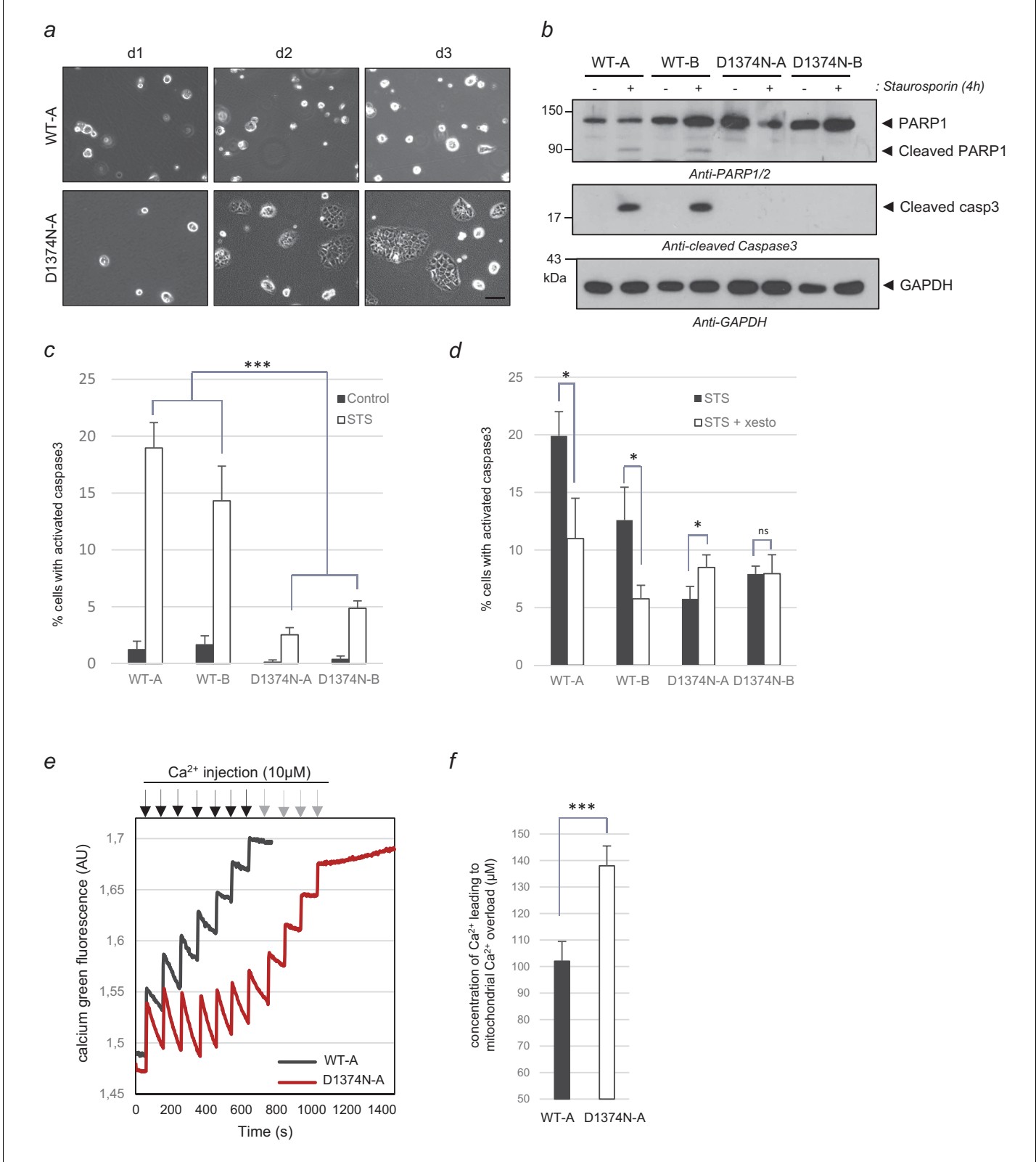

**Figure 5.** Comparison of WT and MET D1374N BMEL cell death. (a) Representative pictures of BMEL cells (clones WT-A and D1374N-A) cultured for 1, 2, and 3 days (d1, d2 and d3) without a collagen coating; scale bar = 100 μm. (b–c) BMEL cells were cultured for 24 hr on a (b) Petri dish or (c) an ibidi slide, both coated with poly-L-lysine, and treated for 4 hr with 1 μM staurosporine. (b) For each condition, the same amount of whole-cell lysate was analyzed by western blotting with antibodies against cleaved caspase 3, PARP1, and GAPDH. (c) Cells were fixed and immunofluorescence staining was

*Figure 5 continued on next page*

*Figure 5 continued*

performed with anti-cleaved caspase 3 antibody. Nuclei were stained with Hoechst. At least 200 cells per well were counted and the percentages of cleaved-caspase 3 positive cells, averaged over three independent experiments, are represented ($n$ = 6;± S.D.). (**d**) BMEL cells were cultured for 24 hr on an ibidi slide coated with poly-L-lysine and treated for 4 hr with 5 µM xestospongin-B and 1 µM staurosporine. Immunofluorescence staining was performed with an anti-cleaved caspase-3 antibody. The nuclei were stained with Hoechst. At least 80 cells per well were counted and percentages of cells displaying cleaved caspase 3 are shown ($n$ = 3;± S.D.) (**e**) BMEL cells (WT-A and D1374N-A) were treated for 4 hr with 1 µM staurosporine to induce apoptosis. The mitochondrial $Ca^{2+}$ uptake capacity of digitonin-permeabilized BMEL cells (250000/ml) was measured with a cytosolic calcium green $Ca^{2+}$ probe upon addition of sequential $Ca^{2+}$ pulses (black and gray arrows) to the medium in an O2K-oxygraph apparatus (Oroboros). (**f**) The measurements of three independent experiments were averaged. ns, nonsignificant; *, $p<0.05$; **, $p<0.01$; ***, $p<0.001$ as determined by Student's $t$ test. Black arrows = $Ca^{2+}$ injection for WT and D1374N cells; gray arrows = $Ca^{2+}$ injection for D1374N cells only.

The online version of this article includes the following source data and figure supplement(s) for figure 5:

**Source data 1.** Source data of *Figure 5c–d* reporting counting of GFP and active caspase 3 positive cells, computation of the percentage, mean and SD, diagram conception and statistical analyses; source data of *Figure 5f* reporting concentration of Ca++ uptake, computation of the mean and statistical analysis.

**Figure supplement 1.** Culture of BMEL WT-B and D1374N-B clones without collagen coating.

**Figure supplement 2.** Induction of BMEL apoptosis by ABT 737.

significantly greater proportion of mice with a plasma level of transaminases at least doubled (>100% increase over the initial level) with respect to MET D1374N mice (*Figure 6f*). Four hours after FAS injection, 82% of WT mice (14/17) had doubled their plasma ALAT levels vs only 46% (5/13) of MET D1374N mice (*Figure 6f*; *Figure 6—source data 1*). Similarly, the plasma ASAT levels had doubled for all WT mice (17/17) vs 70% of MET D1374N (9/13) mice. This was mirrored by a greater proportion of MET D1374N mice in groups with moderate ALAT and ASAT level increases (<50% increase and 50% to 100% increase) (*Figure 6f*; *Figure 6—source data 1*). The MET D1374N mice thus displayed resistance to FAS-induced apoptosis, which demonstrates the importance of MET C-terminal caspase cleavage for optimal apoptosis in vivo and highlights MET function as a dependence receptor.

## Discussion

The function of a protein is largely determined by its location and by its molecular partners. This is well illustrated by the MET dependence receptor. Whereas full-length MET is located at the plasma membrane and allows response to its ligand, MET cleavage by caspases in the absence of its ligand unleashes an intracellular fragment that can amplify apoptosis. The p40MET fragment thus affects the activity of the mitochondria, a central hub in determining the cell life/death balance.

In this study we demonstrate that p40MET localizes to MAMs, a microdomain where the ER is closely apposed with mitochondria. A growing body of evidence demonstrates that this microdomain is involved in various cellular processes, including intra-organelle $Ca^{2+}$ exchange. The juxtaposed membranes promote ER-mitochondrion $Ca^{2+}$ transfer through the IP3R channel at the ER membrane and through the VDAC and MCU channels at the outer and inner mitochondrial membranes. Massive calcium entry into the mitochondria leads to mPTP opening, enabling the release of mitochondrial components. Interestingly, BCL-2 family proteins are shown to localize to both the ER and the mitochondria and to control calcium homeostasis (*Scorrano et al., 2003*; *Oakes et al., 2005*). The anti-apoptotic members Bcl-2 and Bcl-XL can interact with IP3R channels and modulate ER $Ca^{2+}$ homeostasis, thus promoting apoptosis resistance (*Li et al., 2007*). Conversely, the pro-apoptotic proteins BAX and BAK can induce massive calcium leakage from the ER, followed by mitochondrial $Ca^{2+}$ accumulation and apoptosis (*Nutt et al., 2002*) Recent findings demonstrate that BAK facilitates calcium transfer by promoting contacts between mitochondria and the ER (*Mebratu et al., 2017*). We have previously reported BAK silencing or Bcl-XL overexpression to reduce p40MET-induced apoptosis (*Lefebvre et al., 2013*). Interestingly, the results of our FRET experiments show that p40MET interacts with BAK but not Bcl-XL.

Given the observed localization and BAK being a partner of p40MET, we investigated p40MET possible involvement on calcium flux regulation. We show here that inhibiting calcium exchanges prevents p40MET-induced apoptosis, since culturing cells in $Ca^{2+}$ depleted medium impairs the pro-apoptotic action of p40MET, as does a treatment with an inhibitor of ER (IP3R) or the silencing of

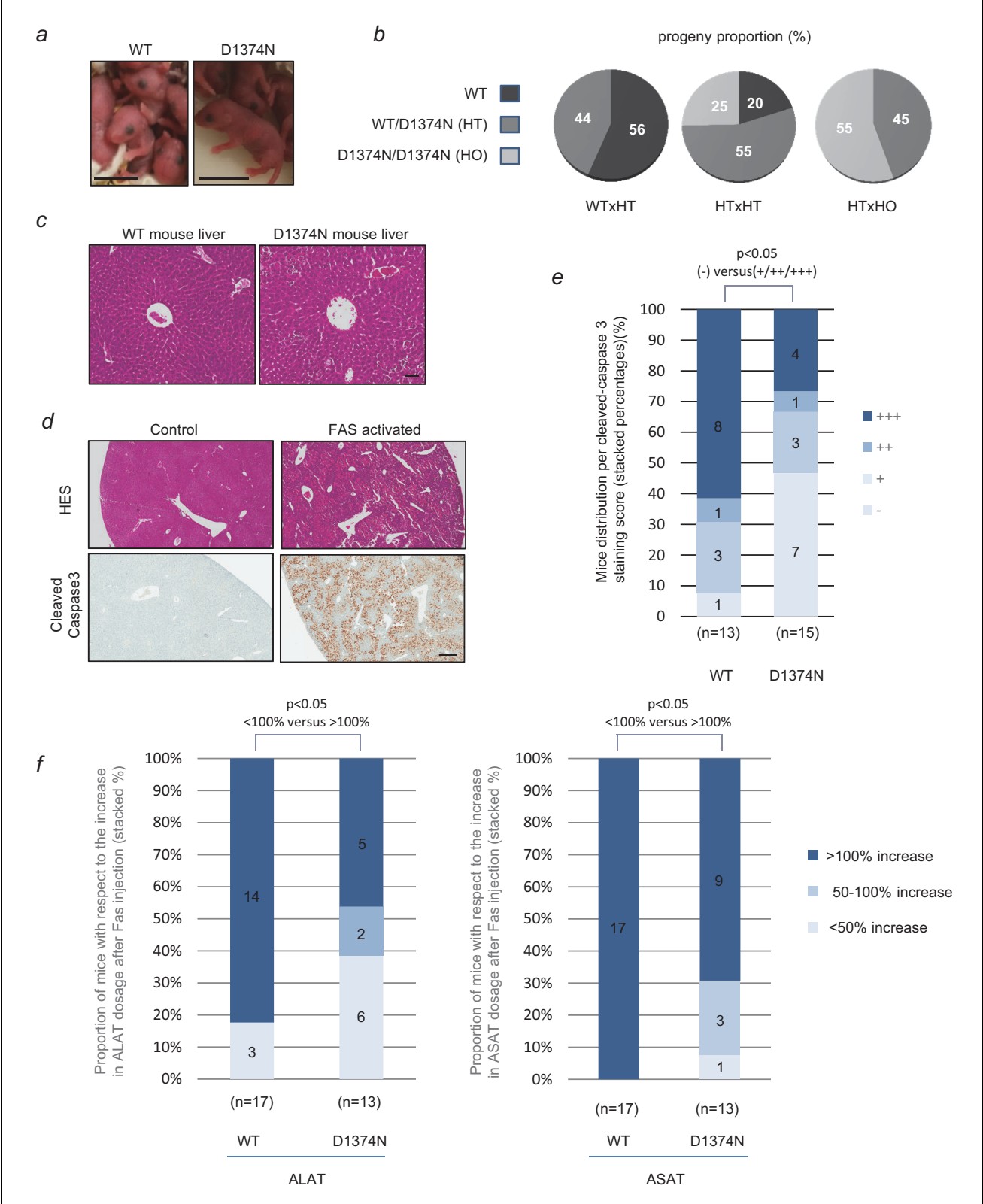

**Figure 6.** Evaluation of hepatocytes apoptosis in WT and D1374N mice. (**a**) Pictures of newborn wild-type (WT) and MET D1374N mice; scale bar = 1 cm. (**b**) Pie chart representation of the proportions of WT, heterozygous (HT) and homozygous (HO) progeny obtained by crossing mice with different genotypes. (**c**) WT and MET D1374N mouse liver sections stained with hematoxylin/eosin, showing a hepatic lobule with a centro-lobular vein at the center; scale bar = 50 µm. (**d**) Example of hematoxylin/eosin and cleaved-caspase 3 staining (brown) in the livers of WT mice pretreated before the

*Figure 6 continued on next page*

*Figure 6 continued*

experiment with 100 µl of 5 mg/ml Crizotinib and then treated for 4 hr with Jo-2 antibody against FAS (4 µg/20 g mouse weight); scale bar = 300 µm. (e) WT and D1374N mice were treated as reported in (d). The total liver area and cleaved caspase-3 staining area were quantified for each slide. The data represent the distribution of cleaved-caspase 3 staining scores (-: 0% to 2%; +: 2% to 5%; ++: 5% to 10%; +++:>10%). In each column, the number of mice obtaining each score is indicated. Statistical analysis applied to differences in negative (-) and positive (+, ++ and +++) staining between WT and D1374N. $p < 0.05$ was determined by Fisher test (WT: $n = 13$; D1374N: $n = 15$). (f) WT and D1374N mice were treated as reported in (d). Blood samples were collected just before and 4 hr after Jo-2 antibody treatment. The graphs represent the distribution of the evolution of plasma ALAT (left) or ASAT (right) levels between these 2 time points (up to 50% increase, 50–100% increase or more than 100% increase). Statistical analysis applied to differences in <100% increase and >100% increase between WT and D1374N. $p < 0.05$ was determined by Fisher test (WT: $n = 17$; D1374N: $n = 13$). The online version of this article includes the following source data and figure supplement(s) for figure 6:

**Source data 1.** Source data of *Figure 6e* reporting percentage of active caspase 3 in liver IHC, repartition in staining score (-;+;++;+++), computation of the percentages, diagram conception and statistical analyses; source data of *Figure 6f* reporting ALAT and ASAT concentration in mouse blood, relative increase, diagram conception and statistical analyses.
**Figure supplement 1.** Mammary gland organization in WT and MET D1374N mice.
**Figure supplement 2.** Evidence of Crizotinib efficacy against HGF-induced survival in BMEL cells.
**Figure supplement 3.** Complete-panel illustration of cleaved-caspase 3 staining of liver slices.

mitochondrial (MCU) calcium channels. Furthermore, using $Ca^{2+}$ probes with different specificities, we have shown that in the presence of p40MET, the ER is depleted of $Ca^{2+}$ while the mitochondria are overloaded. The MAM location of p40MET is thus in agreement with its action on the intrinsic apoptotic pathway. In the light of our various findings, we propose that upon MET cleavage by caspases, p40MET is released into the cytoplasm where it interacts with BAK at MAMs and promotes a $Ca^{2+}$ flux from the ER to the mitochondria, causing mitochondrial permeabilization and apoptosis amplification (*Figure 7*).

The MET receptor's ability to induce both survival in the presence of ligand and apoptosis in the absence of ligand enables us to classify it as a dependence receptor. Dependence receptors displayed a great variety of mechanisms to promote apoptosis (*Negulescu and Mehlen, 2018*). For example, both DCC and UNC5H receptors undergo caspase cleavage and act as caspase activators though death-domain unmasked by the cleavage and carried by the part that remains membrane-anchored (*Mehlen et al., 1998*; *Llambi et al., 2001*). The TrkC receptor, on the other hand, releases upon caspase cleavage a pro-apoptotic fragment which shuttles to the mitochondria and promotes BAX activation (*Ichim et al., 2013*). We highlight here another original mechanism. To the best of our knowledge, this is the first time a dependence receptor has been shown to act via regulation of the $Ca^{2+}$ flux.

Knock-out of the *Met* or *Hgf/sf* gene is lethal in utero and leads to a reduced liver size associated with decreased hepatocyte proliferation (*Schmidt et al., 1995*; *Uehara et al., 1995*). Furthermore, altered liver regeneration has been found in MET-deficient mice (*Huh et al., 2004*). Here we have engineered a novel knock-in model to investigate the physiological impact of the pro-apoptotic p40MET fragment. We have found that liver and mammary gland organogenesis, both of which depend on MET pro-survival signaling, are not affected by the MET D1374N mutation. These data validate the specificity of our model for studying MET pro-apoptotic activity. Following apoptosis induction with a FAS agonist and pretreatment with MET inhibitor to prevent ligand-dependent survival, we observed more apoptosis in WT than in MET D1374N mouse livers, thus evidencing for the first time that MET acts as a dependence receptor in vivo. It would be interesting now to assess the importance of p40MET pro-apoptotic activity in pathological situations such as hepatic steatosis, in which cells expressing the MET receptor face and try to adapt to stresses.

Dependence receptors have placed the role of apoptosis in tumorigenesis in an unexpected perspective. Overexpression of the ligand of the Deleted in Colorectal Cancer (DCC) dependence receptor inhibits its pro-apoptotic activity and can promote spontaneous adenoma formation (*Mazelin et al., 2004*). Although DCC-deficient mice (like MET D1374N mice, data not shown) do not develop spontaneous tumors, experiments in which they were crossed with mice bearing a cancer-predisposing APC mutation suggest that DCC acts as a conditional tumor suppressor gene (*Castets et al., 2012*). In agreement with this concept, the COSMIC (Catalogue of Somatic Mutations in Cancer) database lists six MET mutations affecting the MET caspase sites in various cancers. Also described are exon 14 splice-site mutations found in 3% of pulmonary tumors and leading to

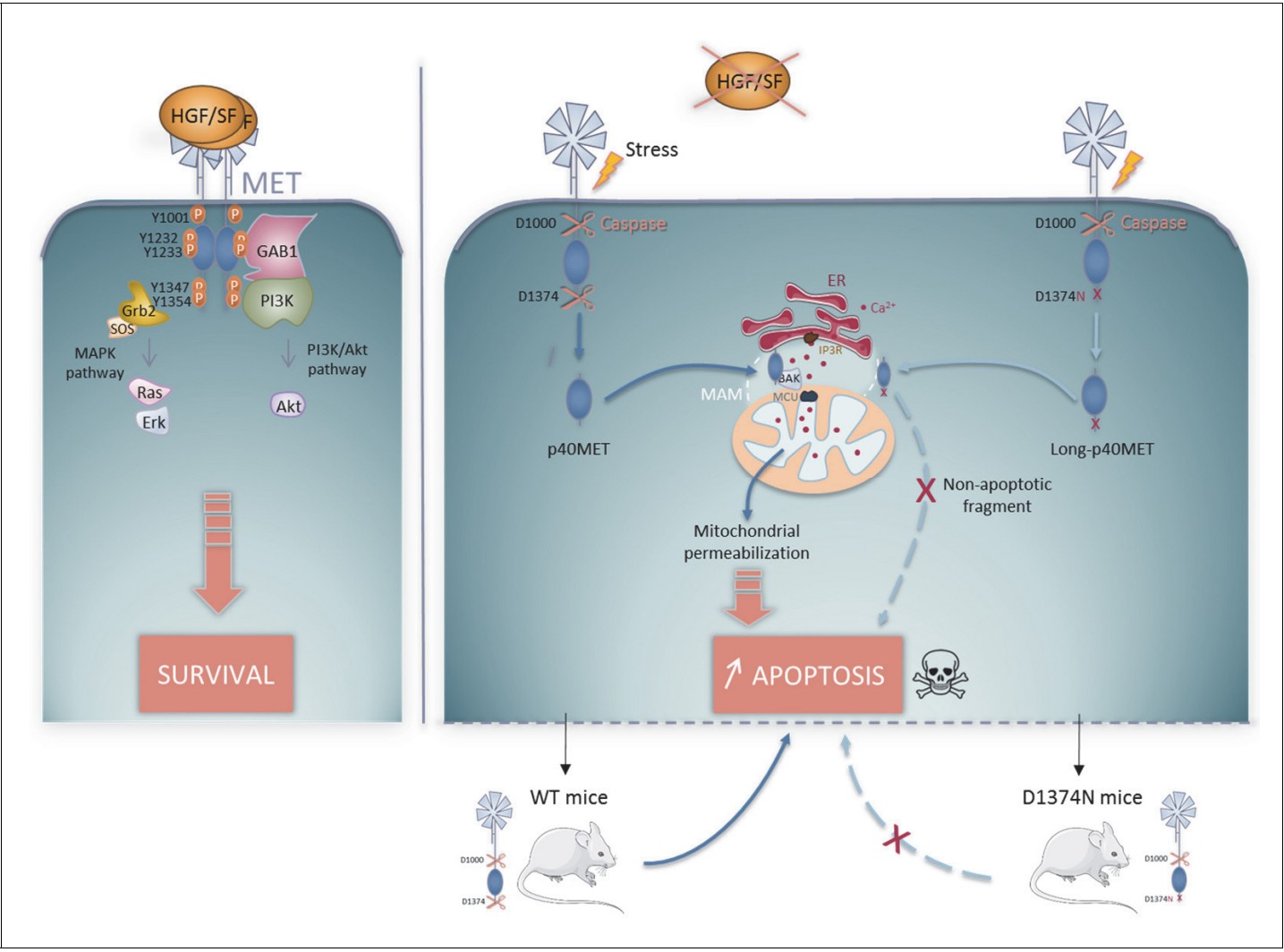

**Figure 7.** Schematic representation of the MET pro-survival and pro-apoptotic pathways in the presence of HGF/SF, MET receptor dimerizes and activates pro-survival signaling by activating the MAPK and PI3K/AKT pathways. In the absence of ligand, MET is cleaved by caspase 3 at a juxtamembrane and a C-terminal site to generate the p40MET fragment, which translocates to the MAM microdomain. p40MET may interact with BAK and promote deregulation of the $Ca^{2+}$ flux between the ER and the mitochondria, causing mitochondrial permeabilization involved in amplification of apoptosis. This dual role of MET classifies it as a dependence receptor. D1374N MET, mutated at the C-terminal caspase site, generates a slightly longer fragment that can no longer promote apoptosis. Upon stress induction by FAS activation in WT and D1374N mice, D1374N hepatocytes show less apoptosis. This suggests that C-terminal caspase cleavage of MET is important for optimal apoptosis in vivo.

loss of the MET juxtamembrane domain, which contains the caspase site leading to p40MET release (**Duplaquet et al., 2018**) Altogether, the presence of these multiple MET-caspase-cleavage-abolishing mutations supports the notion that the MET pro-apoptotic function might put a brake on tumorigenesis.

This study highlights MET as a key regulator of the cell survival-apoptosis balance. In addition to shedding light on pathophysiological mechanisms, this feature of MET may be relevant to their therapeutic targeting. In diseases with exacerbated cell death, such as hepatic steatosis, one might restore the balance by favoring MET pro-survival signaling with ectopic HGF/SF treatment. To favor cancer cell death on the other hand, one might combine MET kinase activity inhibition with agents promoting its pro-apoptotic action. Elucidating the mechanisms of MET pro-survival and pro-death signaling is instrumental to identifying new possible targets and to expanding, potentially, the therapeutic arsenal.

## Materials and methods

### Cytokines, drugs, and cell cultures

HGF/SF was purchased from Peprotech (Rocky Hill, NJ, USA) Anisomycin from Calbiochem (San Diego, CA, USA), staurosporine from Sigma (St Louis, MO, USA) and ABT 737 from Santa Cruz Biotechnology (Santa Cruz, CA, USA). Pan-caspase inhibitor zVAD-FMK was purchased from Calbiochem and Q-VD-OPH from Sigma. HEK293 cells (CRL-11268) and MCF10A human mammary epithelial cells (CRL-10317) were purchased at the ATCC and were cultured as previously described (*Lefebvre et al., 2013*). Bipotential mouse embryonic liver (BMEL) cells were harvested from E14 embryos as described (*Strick-Marchand and Weiss, 2002*). The immortalized human hepatocytes (IHH) cells were provided by Dr F. Kuipers (Groningen, The Netherlands) and cultured as previously described (*Schippers et al., 1997*). All our cell lines were cultured in presence of 1% ZellShield (Minerva Biolabs) and are tested every three months for presence of mycoplasma (MycoAlert LT07-218, Lonza, ME, USA).

### Transfections and RNA interference

Transfections of HEK293 cells on 6-well plates and 60 mm Petri dishes were performed with FuGENE HD (Promega; 2 µg DNA, 8 µl FuGENE, 100 µl Opti-MEM in 1.5 ml complete medium) or in X-tremeGENE 9 DNA (Roche; 2 µg DNA, 3 µl X-tremeGENE 9, 100 µl medium without serum in 3 ml complete medium). For the immunofluorescence, MCF10A or IHH were plated on glass coverslips in 12-well plates at 100,000 and 50,000 cells per well, respectively. The next day, they were transfected with FuGENE HD. For MCU, BAK, BAX and BOK silencing, MCF10A or IHH were first transfected with Lipofectamine 2000 (2.5 µl per ml final volume) with control Stealth siRNA or the following siRNA; for MCU (a combination of three siRNAs: HSS132001-3 Invitrogen; 150 nM final concentration); for BAK (HSS184085; 50 nM); for BAX (HSS141354; 50 nM); for BOK (HSS141392; 50 nM). After that, transfection with the plasmid was carried out as described. Analyses were performed 24 hr or 48 hr after plasmid transfection for MCF10A and IHH, respectively. The previously described MET siRNA (*Kherrouche et al., 2015*) were transfected in IHH cells cultured in 6-well plates at 150,000 cells/well as described above. For FRET, HEK293T cells were seeded on glass-bottom dishes (MatTek; coverslip #1.5) and transfected in FuGENE HD.

### Plasmid engineering

p40MET-EGFP constructs were obtained by insertion of the PCR-amplified cDNA with primers containing XhoI and BamHI restriction sites into pEGFPC3 (Clontech). The p40MET mutated at L1110E and D1115E was created with the QuickChange site-directed mutagenesis system of Stratagene. BAK, BAX, BOK, and BCL-XL fusions with mTurquoise 2 and SYFP2 were generated by PCR amplification on Flag-tagged BAK, BAX or BCL-XL or BOK-EGFP (kind gift from Pr Thomas Kaufmann, University of Bern, Switzerland), with EcoRI and XhoI restriction enzymes into mTurquoise2 and SYFP2 plasmids (Pr T.W. J. Gadella and Dr. J. Goedhart, Amsterdam). p40MET-mTurquoise 2 and SYFP2 were obtained by PCR amplification of p40MET-EGFP and cloned with KpnI and BamHI. The IP3R1-EYFP plasmid was a kind gift from Dr. Geert Bultynck, Leuven, Belgium.

### Antibodies

FAS antibody (Jo-2) was purchased from Becton Dickinson. Antibody against the MET cytoplasmic domain was purchased from Life Technology (3D4/37-0100). Antibody against the C-terminal domain of human MET (L41G3), the MET phosphorylated tyrosine (Y1234/1235)(#3126), the phospho-ERK (Thr202/Tyr204)(#9106), phospho-AKT (Ser-473) (#9271), MCU, and cleaved Casp3 (asp175, #9661) were purchased from Cell Signaling Technology (Danvers, MA). Antibody against cytochrome C(20E8) was purchased from BD Biosciences (San Jose, CA). Antibody against calnexin (ab75801) was purchased from Abcam (Cambridge, MA), antibody against FACL4/ACSL4 (NBP2-16401) from Novus Biologicals (Oakville, ON, Canada), antibody against GFP from Sigma (Saint Louis, MO). Green-fluorescent Alexa fluor 488 conjugated anti-mouse IgG and red-fluorescent Alexa fluor 594 conjugated anti-rabbit IgG were purchased from Invitrogen. Antibodies against PARP-1 (sc-7150), GAPDH (sc-32233, ERK2 (sc-154) and AKT (sc-8312) were purchased from Santa Cruz

Biotechnology (Santa Cruz, CA). Peroxidase-coupled secondary antibodies were from Jackson Immunoresearch Laboratories (West-Grove, PA).

## Immunofluorescence staining

MCF-10A were plated on glass coverslips in 12-well plates (100,000 per well) and transiently transfected the next day as described. BMEL cells were plated on μ-slide 8-well poly-L-lysine (Ibidi, 20,000 per well) and treated for 4 hr with staurosporine (1 μM). IHH cells were plated on glass coverslips coated with poly-L-lysine (100 μg/ml) in 12-well plates (50,000 per well) and transiently transfected the next day as described. After transfection or treatment, the cells were washed and fixed in 4% PFA. They were permeabilized with PBS containing 0.5% Triton X-100 and blocked 30 min in 0.2% casein. Incubation with primary antibodies was carried out for 1 hr. The cells were washed with PBS and incubated for 1 hr with a combination of Alexa Fluor-conjugated secondary antibodies (Alexa Fluor 488 anti-mouse IgG and Alexa Fluor 594 anti-rabbit IgG; 2 μg/ml). The nuclei were counterstained with Hoechst 33258. For fluorescence microscopy, slides were observed in an Axion Imager Z1 (Carl Zeiss), numerical aperture PC-Plan NEOFLUAR 40x/1.3 oil and the ZEN acquisition software. Ibidi chambered coverslips were observed in an inverted-lens microscope (AxioObserver Z1 Video-DG4 numerical aperture; Plan APOCHROMAT 40x/1.3 oil). Mitochondria were stained with 100 nM MitoTracker (Invitrogen). Cells were washed and fixed in methanol:acetone (1:1 v/v). MAM and ER staining was performed as described. Slides were observed in an LSM 880 Laser Scanning Confocal Microscope (Carl Zeiss), numerical aperture PLAN-APOCHROMAT 63x NA 1.4. Weighted colocalization coefficients were measured with the JACoP plugin of ImageJ software using Manders Coefficients. The immunofluorescence experiments were performed at least two times.

## Immunohistochemistry staining

Livers were fixed in 4% PFA, dehydrated in successive baths (30%, 70%, 95%, and 100% ethanol and toluene), and paraffin-embedded. Immunohistochemistry was performed to detect cleaved caspase 3 using a Ventana Discovery XT autostainer on 4 μm-thick sections. Slides were deparaffinized with EZPrep buffer and epitopes unmasked in CC1 EDTA buffer. Sections were incubated 40 min with anti-cleaved caspase 3 antibody (1/1000). Secondary antibody (Omnimap Rabbit Ventana) was incubated 16 min. After washes, staining was performed with DAB and sections were counterstained with Hematoxylin. Whole slide images were digitized at 20 × using the ScanScope CS scanner (Leica Biosystems, Nussloch, Germany).

## Subcellular fractionation

MCF10A were treated for 4 hr with staurosporine (1 μM) and the MAM, ER, and mitochondrial fractions were isolated as previously described *Wieckowski et al. (2009)*.

## FLIM-FRET

HEK293T cells were seeded into 35 mm glass-bottom dishes (MatTek, Ashland, MA) and transfected as described. Imaging was performed at 37°C in a thermostatic chamber in L-15 medium. FLIM was carried out with a Nikon A1 inverted confocal coordinated with a 440 nm pulsed laser (PicoQuant, Berlin, Germany) set at 40 MHz. FRET experiment were performed by Time-Correlated Single Photon Counting (TCSPC) methods by using hybrid detector (Picoquant) and a TCSPC counting card for photon counting (HydraHarp400; PicoQuant). NIS (Nikon) and SymPhoTime64 (PicoQuant) softwares were used to handle acquisition. Fluorescence lifetime determinations were performed as described *Leray et al. (2013)*. FRET efficiency was calculated as follows: $EFRET = 1\ T_{DA}/T_D$, from measurement of the donor lifetime in presence of the acceptor ($T_{DA}$) and the donor alone lifetime ($T_D$) in reference.

## Calcium measurements

HEK293 cells were grown on a glass bottom dish and transfected as described. 24 hr later, cells were loaded for 30 min at 37°C with 5 μM Fura-2 AM immediately prior to acquisition and rinsed in HBSS medium with 0.04 g EGTA pH >7. The emitted fluorescence of Fura-2 was captured at 510 nm with a photomultiplier-based system (Photon Technologies International Ltd, Princeton, NJ) after cell excitation at alternately 340 and 380 nm. Thapsigargin (1 μM) was added 5 min after the start of

acquisition. [Ca$^{2+}$] was calculated from the ratio of the emitted fluorescence excited by 340 light to that excited by 380 nm light (*Grynkiewicz et al., 1985*). The 4mtD3cpv biosensor was a generous gift from Roger Tsien (University of California, San Diego). FRET transfer was measured by lifetime analysis, as described. To measure mitochondrial calcium in BMEL, the cells were resuspended in calcium buffer (150 mM KCl, 5 mM KH$_2$PO$_4$, 1 mM MgCl$_2$, 5 mM Tris pH 7.4) containing 0.5 µM rotenone, 10 mM succinate, and 2.5 mM ADP. They were then permeabilized with 2.5 µg/millions-of-cell digitonin. Cells were placed in an Oroboros chamber (Oroboros Oxygraph-2k Instruments) and calcium green probe (Life Technologies) was added at 0.2 µM and the fluorescence intensity was measured with an O2K-Fluo Led2-module. A pulse of calcium was delivered every 2 min (10 µM final concentration) until the mitochondria became overloaded.

## Real-time RT-PCR

Total RNA was extracted with the Nucleospin RNA/Protein Kit (Macherey-Nagel). cDNA was reverse-transcribed with random hexamers (Applied Biosystems, Green Island, NY, USA). Levels of *Krt19* (CK19), *Hnf4a* (HNF4a), *Aldob* (aldolase b) and *Alb* (albumin) mRNAs were evaluated by real-time RT-PCR with Fast SYBR Green mix (Applied Biosystems). Relative gene expression levels were calculated using the 2-ΔΔCt method. The level of each transcript was normalized to the *Ppia* (peptidylprolyl isomerase A). The primers used were: *Krt19*, 5'-CCTGGAGATGCAGATTGAGAG-3'; 5'-AGGATCTTGGCTAGGTCGACA-3'); *Hnf4a* (5'-CCTCTTCCTTCCTGGTGC-3'; 5'-GCCTCCAGA-GAGATGGCTTTA-3'), *Aldob* (5'- AGCGGGCTATGGCTAAC-3'; 5'-GAGGCTGTGAAGAGCGAC-3'); and *Alb* (5'- AAGGCTGCTGACAAGGAC-3'; 5'-GGTTGTGGTTGTGATGTG-3').

## Mouse model strategy

Transgenic MET D1374N mice were obtained at the Mouse Clinical Institute in Strasbourg, France by homologous recombination in C57/bl6 embryonic stem cells with a plasmid bearing exons 21, 22 of *Met* gene with the GAT codon coding residue Asp1374 mutated to AAT coding for an Asn. A floxed neo cassette was inserted downstream of the *met* sequence. After Cre recombinase action, a unique loxP site remained. Chimeric mice were obtained after injection of mutated ES cells and crossed with C57/bl6 mice. The mice were cared in accordance with FELASA recommendations with protocols approved by the CEEA75 Ethics Committee (agreement B59-350009). Mice were genotyped by PCR: 5'-AAATCGGTAGCTCTCCGTAATTCATCC-3' and 5'-CCTGAATCAGGCATCTCA-CAATGATCT-3'.

## FAS-injection-induced apoptosis in mice

For the in vivo apoptotic studies, each genotype of male C57BL/6 mice of comparable ages and weighing 20–30 g, were randomly separated and force-fed with 200 µl of 5 mg/ml Crizotinib (Sigma) 5 days, 2 days, and 1 day before the experiment. After fasting overnight, the mice were injected intraperitoneally with anti-FAS antibody (Jo2, 4 µg/20 g). Four hours later, the mice were sacrificed by cerebral dislocation. Their livers were perfused with PBS, quickly removed, and placed in 4% PFA. Plasma aspartate aminotransferase (ASAT) and alanine aminotransferase (ALAT) activities were determined by colorimetric assay following the manufacturer recommendations (Biolabo).

## Western blotting

Western blotting were performed at least two times as previously described *Paumelle et al. (2002)*. For Key Resources Table see the *Supplementary file 1*.

## Acknowledgements

We thank Dr Jérome Kluza, Pr Tony Lefebvre, and all members of our team for help and discussions. We are most grateful to Dr Damien Grégoire and Dr Urszula Hibner for their help in establishing and characterizing BMEL cells. We thank Flavio Maina's team for kindly providing us with murine hepatocyte RNAs. We thank the BioImaging Center Lille (BICeL) and EquipEX ImaginEx BioMed for microscopy supplies and advice. We thank A Leray for the MAPI software. We would like to thank Nathalie Hennuyer for her technical assistance and Ludovic Mercier and David Hannebique for their assistance in maintaining mouse strains.

## Additional information

### Funding

| Funder | Grant reference number | Author |
|---|---|---|
| Institut National Du Cancer | | David Tulasne |
| Cancéropôle Nord-Ouest | | David Tulasne |
| Région Hauts-de-France | CPER Photonics | Alessandro Furlan |
| Ligue Contre le Cancer | | David Tulasne |
| Institut National Du Cancer | SIRIC OncoLille | David Tulasne |

The funders had no role in study design, data collection and interpretation, or the decision to submit the work for publication.

### Author contributions

Leslie Duplaquet, Conceptualization, Formal analysis, Investigation, Methodology, Writing - original draft, Writing - review and editing; Catherine Leroy, Audrey Vinchent, Sonia Paget, Florence Giffard, Formal analysis, Investigation, Methodology; Jonathan Lefebvre, Conceptualization, Formal analysis, Investigation, Methodology; Fabien Vanden Abeele, Gabriel Bidaux, Conceptualization, Investigation, Methodology; Steve Lancel, Conceptualization, Formal analysis, Supervision, Investigation, Methodology; Réjane Paumelle, Conceptualization, Formal analysis, Supervision; Laurent Heliot, Conceptualization, Supervision, Methodology; Laurent Poulain, Conceptualization, Supervision, Investigation; Alessandro Furlan, Conceptualization, Formal analysis, Supervision, Funding acquisition, Validation, Investigation, Visualization, Methodology, Writing - original draft, Writing - review and editing; David Tulasne, Conceptualization, Formal analysis, Supervision, Funding acquisition, Validation, Investigation, Methodology, Project administration, Writing - review and editing

### Author ORCIDs

Leslie Duplaquet https://orcid.org/0000-0002-6544-0432
Fabien Vanden Abeele http://orcid.org/0000-0001-7111-7220
Steve Lancel http://orcid.org/0000-0002-3292-5433
Gabriel Bidaux http://orcid.org/0000-0002-6162-3223
Alessandro Furlan https://orcid.org/0000-0001-6502-8416
David Tulasne https://orcid.org/0000-0002-6764-7242

### Ethics

Animal experimentation: The mice were cared in accordance with FELASA recommendations with protocols approved by the CEEA75 Ethics Committee (agreement B59-350009).

### Decision letter and Author response

Decision letter https://doi.org/10.7554/eLife.50041.sa1
Author response https://doi.org/10.7554/eLife.50041.sa2

## Additional files

### Supplementary files

- Supplementary file 1. Key Resources Table.
- Transparent reporting form

### Data availability

All data generated or analysed during this study are included in the manuscript and supporting files.

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
