## [Decision Letter]

**Acceptance summary:**

This manuscript demonstrates that a fragment of the receptor tyrosine kinase MET (p40MET) can localize to microdomains of contact between the endoplasmic reticulum and mitochondria, where it interacts with the pro-apoptotic pore-forming molecule BAK to mediate Fas-induced apoptosis in an in vivo model of fulminant hepatitis. This is a novel function and mechanism for this protein, with relevance to human pathology.

**Decision letter after peer review:**

Thank you for submitting your article "Control of cell death/survival balance by the MET dependence receptor" for consideration by *eLife*. Your article has been reviewed by two peer reviewers, one of whom is a member of our Board of Reviewing Editors, and the evaluation has been overseen by Jonathan Cooper as the Senior Editor. The reviewers have opted to remain anonymous.

The reviewers have discussed the reviews with one another and the Reviewing Editor has drafted this decision to help you prepare a revised submission.

Summary:

The current manuscript by Duplaquet et al. builds upon previous work by this same team to investigate the mechanism by which the caspase-cleavage product of the receptor tyrosine kinase MET (p40MET) induces apoptosis.

New important observations in this manuscript are that p40MET localized to microdomains of contact between the ER and mitochondria (MAMs), that p40MET interacts with the pro-poapoptotic pore-forming molecule BAK, that calcium flux from the ER to the mitochondria is required for p40MET-induced apoptosis, and most interestingly, that p40MET is required for Fas-induced apoptosis in an in vivo model of fulminant hepatitis.

Overall, the manuscript is well written and describes well supported conclusions. The in vivo model is particularly important to demonstrate a physiological role for p40MET in apoptosis.

However, the manuscript falls short in terms of mechanistic insights and could be significantly enhanced by a deeper understanding of how p40MET contributes to the apoptotic cascade.

Essential revisions:

1) In the interaction studies in Figure 2, p40MET is shown to interact with BAK, but not with the pro-survival BH3 protein BCL-XL. However, there is no data in the manuscript demonstrating that BAK is required for cytochrome C release, calcium efflux or caspase activation in response to p40MET overexpression. To gain further insight, the authors should test for interaction of p40MET with other pore forming proteins in the FRET assays, such as BAX and BOK. They should then test which if any of these proteins are required for p40MET-induced cytochrome C release and apoptosis.

2) In their in vivo experiments, they focused on the function of MET in the liver survival‐apoptosis balance, but their in vitro mechanistic experiments employed HEK293 cells (kidney epithelial cell line) and MCF10A (mammary epithelial cell lines). The reviewers request that key mechanistic experiments, such as localization to MAMs, interaction with pore-forming proteins, and apoptotic assays, be repeated in a cell line of hepatic origin.

---

## [Author Response]

Essential revisions:1) In the interaction studies in Figure 2, p40MET is shown to interact with BAK, but not with the pro-survival BH3 protein BCL-XL. However, there is no data in the manuscript demonstrating that BAK is required for cytochrome C release, calcium efflux or caspase activation in response to p40MET overexpression. To gain further insight, the authors should test for interaction of p40MET with other pore forming proteins in the FRET assays, such as BAX and BOK. They should then test which if any of these proteins are required for p40MET-induced cytochrome C release and apoptosis.

We previously demonstrated that the apoptosis triggered by the transfection of p40MET-flag in MCF10A cells was dependent on BAK, as evidenced by the decrease of cytochrome C when BAK was silenced (Lefebvre et al., 2013).

To reinforce these functional data, MCF10A were transfected with p40MET and with siRNA against BAK, BAX or BOK. BAK siRNA abrogated MET fragment induced caspase 3 activation and cytochrome C release, BOK silencing partially inhibited them, while BAX silencing had no effect (novel Figure 2—figure supplement 1). Similar results were obtained in IHH cells regarding p40MET induced caspase 3 activation (novel Figure 2—figure supplement 2). It is worth noticing that cytochrome C release was hardly visible in IHH cells probably due to small volume of the cytoplasm compared to the nucleus Taken together, these results confirmed the main involvement of BAK in p40MET-induced apoptosis.

To gain further insight, following the reviewers’ advice, we extended our study to interactions with other pore forming proteins including BAX and BOK. For that purpose, we built vectors of expression for fusion proteins with fluorescent proteins to be used in FRET experiments. These experiments revealed an interaction of p40MET with BAX (Figure 2—figure supplement 4A-B). Still, as observed in our previous publication with p40MET-FLAG, BAX silencing did not affect the apoptosis induced by p40MET (novel Figure 2—figure supplement 1 and 2).

BOK hardly interacted with p40MET as assessed by FRET (novel Figure 2—figure supplement 4A-C) and its silencing interfered to a certain extent with p40MET-triggered apoptosis (novel Figure 2—figure supplement 1 and 2), possibly via indirect interactions with BAK, which we reported in our FRET experiments (Figure 2—figure supplement 4C).

2) In their in vivo experiments, they focused on the function of MET in the liver survival‐apoptosis balance, but their in vitro mechanistic experiments employed HEK293 cells (kidney epithelial cell line) and MCF10A (mammary epithelial cell lines). The reviewers request that key mechanistic experiments, such as localization to MAMs, interaction with pore-forming proteins, and apoptotic assays, be repeated in a cell line of hepatic origin.

In order to confirm our mechanistic insights in cells of hepatic origin, we repeated key experiments with the IHH (Immortalized Human Hepatocyte) cell line.

In these cells p40MET was efficiently produced, in a caspase-dependent manner, when apoptosis was triggered with BH3 mimetic ABT 737 (novel Figure 1—figure supplement 2A-B). MET siRNA confirmed that the generated 40kDa fragment is indeed produced from MET receptor (novel Figure 1—figure supplement 2C). In addition, transfection of GFP-p40MET in this cell line induced caspase 3 activation, in contrast to the GFP-p40MET D1374N or GFP alone (novel Figure 1D). Treatment with calcium chelator BAPTA-AM abrogated p40MET induced apoptosis in IHH demonstrated that calcium flux regulation is required for MET pro-apoptotic property in cell of hepatic origin (novel Figure 3C).

As previously mentioned, the importance of BAK in this process in IHH cells was supported by FRET experiments (novel Figure 2—figure supplement 4D-E) and by the great reduction of cleaved caspase 3 immunostaining when BAK was silenced by siRNA (novel Figure 2—figure supplement 2).